# PLEKHM1 Overexpression Impairs Autophagy and Exacerbates Neurodegeneration in rAAV-α-Synuclein Mice

**DOI:** 10.3390/cells14171340

**Published:** 2025-08-29

**Authors:** Lennart Höfs, David Geißler-Lösch, Björn H. Falkenburger

**Affiliations:** 1Department of Neurology, Faculty of Medicine, University Hospital Carl Gustav Carus, Technische Universität Dresden, 01307 Dresden, Germany; 2Deutsches Zentrum für Neurodegenerative Erkrankungen (DZNE), 01307 Dresden, Germany

**Keywords:** alpha-synuclein, autophagy, autophagic flux, lysosomes, RFP-EGFP-LC3 mice, mouse model

## Abstract

The aggregation of α-synuclein (αSyn) is a central feature of Parkinson’s disease (PD) and other synucleinopathies. The efficient clearance of αSyn depends largely on the autophagy–lysosomal pathway. Emerging genetic evidence highlights the role of pleckstrin homology and RUN domain-containing M1 protein (PLEKHM1), a critical regulator of autophagosome–lysosome fusion, in the pathogenesis of multiple neurodegenerative diseases. This study investigates the possible effects of increased PLEKHM1 expression on αSyn pathology and neurodegeneration in mice. We utilized a mouse model of PD that is based on A53T-αSyn overexpression, achieved by the stereotactic injection of recombinant adeno-associated viral vectors (rAAV) into the substantia nigra. Additionally, this study explores the effect of PLEKHM1 overexpression on the autophagy–lysosomal pathway under physiological conditions, using transgenic autophagy reporter mice. PLEKHM1 overexpression facilitated the αSyn-induced degeneration of dopaminergic somata in the substantia nigra and degeneration of dopaminergic axon terminals in the striatum. In concert with αSyn expression, PLEKHM1 also potentiated microglial activation. The extent of αSyn pathology, as reported by staining for phosphorylated αSyn, was not affected by PLEKHM1. Using RFP-EGFP-LC3 autophagy reporter mice, rAAV-mediated PLEKHM1 overexpression reduced lysosomal and autolysosomal area, increased LAMP1-LC3 colocalization, and decreased the autolysosome-to-autophagosome ratio. Concurrently, PLEKHM1 overexpression in both genotypes caused p62 accumulation, accompanied by reduced overlap with lysosomal and autophagosomal markers but increased colocalization with autolysosomal markers, indicating impaired cargo degradation during late-stage autophagy. Taken together, elevated PLEKHM1 levels exacerbate neurodegeneration in αSyn-overexpressing mice, possibly by impairing autophagic flux. Now, with in vivo evidence complementing genetic data, alterations in PLEKHM1 expression appear to compromise autophagy, potentially enhancing neuronal vulnerability to secondary insults like αSyn pathology.

## 1. Introduction

Neuronal synucleinopathies such as Parkinson’s disease (PD) and dementia with Lewy bodies (DLB) are neurodegenerative diseases characterized by intracellular aggregates of the synaptic protein α-synuclein (αSyn) [1,2,3]. αSyn aggregates impair intracellular transport, rupture lipid membranes, disrupt autophagic flux, induce neuroinflammation and cause DNA damage [4]. They induce neurodegeneration, particularly affecting dopaminergic neurons. αSyn aggregates are mostly cleared through the autophagy–lysosomal pathway [5,6]. As a consequence, augmenting autolysosome formation to enhance the degradation of toxic misfolded αSyn is considered a potential therapeutic approach [5,7,8].

Several genetic and epigenetic studies implicated the autophagy adapter protein pleckstrin homology and RUN domain-containing M1 (PLEKHM1) in the pathogenesis of neurodegenerative disorders. Genome-wide association studies revealed single nucleotide polymorphisms in the PLEKHM1 gene that increase the risk for Alzheimer’s disease (AD) and PD [9,10]. The expression of PLEKHM1 seems to be intricately regulated, as the expression of PLEKHM1 is increased in the MAPT H1 haplotype, which is associated with AD, PD, progressive supranuclear palsy (PSP), and corticobasal degeneration [11]. Additionally, specific methylation patterns of the PLEKHM1 gene have been negatively associated with PD risk [12]. Conversely, decreased PLEKHM1 expression due to promoter methylation has been identified as a risk factor for PSP [13].

PLEKHM1 was initially characterized for its role in hereditary osteopetrosis, which can be caused by loss-of-function mutations in the PLEKHM1 gene [14,15]. Subsequent studies revealed that PLEKHM1 directly interacts with the microtubule-associated protein 1A/1B-light chain 3 (LC3) and Rab7. By recruiting the homotypic fusion and protein-sorting complex (HOPS complex), PLEKHM1 is involved in the recruitment of SNARE proteins for membrane fusion, and it thereby orchestrates fusion events in the autophagy–lysosomal pathway [16]. The siRNA-mediated depletion of PLEKHM1 in vitro inhibits the lysosomal degradation of endocytic cargo, and genetic knockout of PLEKHM1 in mice inhibits autophagosome–lysosome fusion [17].

Since alterations in PLEKHM1 expression levels, in either direction, were linked to neurodegenerative diseases and the lack of PLEKHM1 impairs the autophagy–lysosomal pathway, we were interested in the impact of PLEKHM1 overexpression. We hypothesized that PLEKHM1 overexpression modulates autophagic clearance—either by facilitating or impairing it—thereby potentially affecting αSyn pathology and providing insights into disease mechanisms and the functional relevance of genetic evidence. To assess whether increased PLEKHM1 levels in mouse neurons affect αSyn-induced toxicity, we utilized a recombinant adeno-associated vector (rAAV) to transduce PLEKHM1. The rAAV was stereotactically injected into the substantia nigra (SN), alongside a separate rAAV overexpressing the pathogenic A53T mutant of human αSyn [18]. The increased expression of monomeric αSyn causes familial variants of PD [19,20]. Additionally, PLEKHM1 was transduced in transgenic autophagy reporter mice to examine the impact of PLEKHM1 overexpression on the autophagy–lysosomal pathway under physiological conditions.

## 2. Materials and Methods

### 2.1. Production of the PLEKHM1 Transfer Plasmid for rAAV Production

The AAV transfer plasmid hPlekhm1-dsRedM, which provides the DNA sequence to be packaged into the rAAV capsid, was a gift from Paul Odgren and obtained from Addgene (Plasmid #73592). The dsRedM ORF at the C-terminus of PLEKHM1 was replaced with a hemagglutinin epitope tag (HA-tag). Incorporating a HA-tag enables straightforward detection of the expressed PLEKHM1 protein. The high-fidelity restriction enzymes NotI-HF (cat. no. R3189, New England Biolabs, Ipswich, MA, USA) and KpnI-HF (cat. no. R3142, New England Biolabs) were used. Restriction fragments were separated by agarose gel electrophoresis and excised from the gel using the QIAquick Gel Extraction Kit (Qiagen, cat. no. 28704, Hilden, Germany). Annealed oligonucleotides encoding the HA tag were phosphorylated at the 5′ end using T4 polynucleotide kinase (cat. no. M0201SVIAL, New England Biolabs). Ligation was performed utilizing T4 DNA ligase (New England Biolabs) in ‘10x 4 Ligation NEB Buffer’ (New England Biolabs). The resulting plasmid was amplified in DH5-alpha bacteria (cat. no. 18263012, Invitrogen, Waltham, MA, USA) under ampicillin selection. Next, the PLEKHM1-HA-tag sequence was cut out from the newly generated plasmid using Nhel-HF (cat. no. 3131, New England Biolabs) and Notl-HF (cat. no. R3189, New England Biolabs). The fragment was inserted into the AAV-backbone plasmid using the Quick Ligation Kit (cat. no. M2200S, New England Biolabs). The final rAAV transfer plasmid contains a CMVie-enhanced synapsin1 promoter, a WPRE sequence and bovine growth hormone polyadenylation sequence flanked by inverted terminal repeats.

### 2.2. rAAV Production

rAAV-encoding A53T-αSyn was generated by the Leuven Viral Vector Core, as described previously [21,22]. rAAVs encoding PLEKHM1 and EGFP were generated at the Viral Core Facility at Charité Universitätsmedizin Berlin. The pseudotyped rAAV2/7 vector was selected for its high efficiency in neuronal transduction. For each rAAV used, genomic copies (GC) were determined as technical titers by real-time PCR analysis. Each rAAV was diluted to a concentration of 3 × 10^12^ GC/mL, and aliquoted and stored at −80 °C until further use as described previously [21,22]. Two microliters of a 1:1 mixture of the viral vectors were stereotactically injected. The rAAV-EGFP control group received only rAAV-EGFP, but the total number of GC matched that used in the other groups.

### 2.3. Animals

All animal experiments were conducted in compliance with EU Directive 2010/63 on the protection of animals used for scientific purposes and were approved by the Animal Research Committee of Landesdirektion Sachsen, Dresden, Germany (25-5131/49615) on 5 May 2020.

All mice were housed under a 12 h light and dark cycle with free access to pellet food and water in the Experimental Center, TU Dresden, Dresden, Germany.

For the first experiment, twelve-week-old C57BLl/6 male mice were obtained from Janvier Labs (Le Genest Saint Isle, France). We included five animals per group, except for the aSyn + PLEKHM1 group, which comprised six animals. The one extra mouse was initially included as a precaution against potential loss but was ultimately not required. Sample sizes were determined based on our prior studies with the goal of minimizing animal use while ensuring sufficient statistical power. In the second experiment involving transgenic mice, six male RFP-EGFP-LC3 transgenic mice were used; initially developed by Li et al. [23] and obtained from the Jackson Laboratories (RRID: IMSR_JAX:027139; Jackson Laboratories, Bar Harbor, ME, USA). In total, 27 mice were included in this study. All animals designated for the experiments were included in the analysis; thus, no specific inclusion or exclusion criteria were applied.

Cages were randomly assigned to receive one of the rAAV treatments. To reduce potential confounding variables, rAAV injections were performed on four consecutive days, thereby minimizing the time difference between key experimental groups. Perfusions were completed within a single day to further control for variability. Staining protocols were standardized, and animals were randomly selected for staining since processing could not be completed in a single session. Imaging parameters were defined once per epitope and remained constant throughout the study. One investigator (D.G.-L.) as well as lab technicians were fully blinded to group assignments during key steps, including the sectioning, staining, imaging, and quantification of dopaminergic neurons. Further image analyses were conducted using automated and independent analysis tools to ensure objectivity (see paragraphs below).

### 2.4. Stereotactic Surgery

Unilateral stereotactic surgeries were conducted under aseptic conditions. Mice were anesthetized with intraperitoneal injections of Ketamine (100 mg/kg) and Xylazine (10 mg/kg), and cooling of the body temperature was prevented using a heated cushion. The animals were positioned in a flat skull orientation within a stereotactic head frame, and a bore-hole craniotomy was performed using the following stereotactic coordinates relative to Bregma: anterior–posterior −3 mm, lateral −1.2 mm and dorso-ventral −4.1 mm. Following a manual incision of the dura, a Hamilton glass syringe (Hamilton Bonaduz AG, Bonaduz, Switzerland) was used to inject 2 µL of the vector solution. The needle was advanced at a rate of 300 µm/min until reaching the final coordinates, where it was paused for five minutes both before and after the vector injection. Two hundred mL viral vector solutions were injected per minute. The syringe was slowly withdrawn and the skin was sutured. Lidocaine (Aspen Pharmacare, Durban, South Africa) was topically applied and Metamizole (WDT, Garbsen, Germany) was added to the drinking water.

### 2.5. Tissue Preparation

Eight weeks after virus injection, animals were sacrificed with an overdose of isoflurane (Baxter, Lessines, Belgium). Subsequently, transcardial perfusion was performed using 4% paraformaldehyde (PFA) which was diluted in Tris-buffered saline (TBS, pH 7.6). For further fixation, the brains were immersed in 4% PFA for 48 h at 4 °C, followed by cryoprotection in TBS containing 30% sucrose. Next, the tissue was frozen in isopentane at −55 °C and stored at −80 °C until it was sectioned into 30 µm thick coronal brain slices using a cryostat (Leica Biosystems, Nussloch, Germany).

### 2.6. Immunohistochemistry and Immunofluorescence Stainings

Coronal brain sections were stained using primary antibodies, fluorophore-linked or enzyme-linked secondary antibodies. The following antigens were visualized: tyrosine hydroxylase (TH), human αSyn (H-αSyn), phosphorylated αSyn at Serine 129 (P-αSyn), glial fibrillary acidic protein (GFAP), ionized calcium-binding adapter molecule 1 (Iba1), lysosomal-associated membrane protein 1 (LAMP1), p62 (SQSTM1), and the HA-tag added to pleckstrin homology and RUN domain-containing protein 1 (PLEKHM1). For immunofluorescence staining, sections were rinsed three times for 10 min in TBS, followed by incubation in a blocking solution for 1 h at room temperature. The blocking solution contained 10% donkey serum (BIOZOL Diagnostica Vertrieb GmbH, Eching, Germany) and 0.2% Triton X-100 (Thermo Scientific, Waltham, MA, USA) in TBS. Subsequently, the sections were incubated for 20 h at 4 °C with combinations of the following primary antibodies: rabbit anti-TH (1:1000, P40101, Pel-Freez, Rogers, AR, USA), rabbit anti-P-αSyn (1:1000, ab51253, Abcam, Cambridge, UK), rat anti-H-αSyn (1:1000, ALX-804-258-L001, ENZO Life Sciences, Farmingdale, NY, USA), chicken anti-GFAP (1:500, ab4674, Abcam), guinea pig anti-Iba1 (1:2000, HS-234308, Histo Sure, Göttingen, Germany), rabbit anti-p62 (1:500, ab56416, Abcam), and LAMP1 (1:600, ab208943, Abcam). After another three 10 min rinses, the sections were treated with fluorophore-conjugated secondary antibodies for 1 h at 21 °C: CF555-conjugated donkey anti-guinea pig (1:1000, SAB4600298, Sigma-Aldrich, St. Louis, MO, USA), Rhodamine red-conjugated donkey anti-rabbit (1:1000, 711-295-152, Jackson ImmunoResearch, Ely, UK), Alexa 647 donkey anti-rat (1:1000, ab150155, Abcam), Alexa 647-conjugated donkey anti-rabbit (1:1000, A31573, Molecular Probes, Eugene, OR, USA), and Alexa 405-conjugated goat anti-chicken (1:500, A48260, Invitrogen). Nuclei were counterstained with Hoechst (1:2000 for 5 min; Invitrogen, Waltham, MA, USA) and the sections were mounted in Fluoromount-G (Invitrogen).

The HA-tagged PLEKHM1 was visualized using an enzyme-linked staining protocol based on the horseradish peroxidase reaction using Vector VIP Peroxidase Substrate (SK-4600; Vector Laboratories, Newark, CA, USA). Three sections of each animal, that was transduced with rAAV-PLEKHM1, were rinsed in TBS for ten minutes. The quenching of endogenous peroxidase was performed in H_2_O_2_ (3%) for 15 min. After additional rinses, the sections were incubated in 10% donkey serum for 1 h. The primary antibody, rabbit anti-HA (1:1000, C29F4, Cell Signaling Technology, Danvers, MA, USA), was added for 18 h at 4 °C. Next, sections were rinsed three times and incubated in HRP-conjugated donkey anti-rabbit secondary antibody (1:1000, 711-035-152, Jackson Immunoresearch). The sections were rinsed again. The reagent solution was prepared according to the manufacturer’s protocol applied onto the sections for 7.5 min. The tissue underwent one more rising cycle, was dehydrated in ethanol and finally mounted in Euparal (Waldek GmbH & Co. KG, Münster, Germany).

### 2.7. Quantification of TH-Positive Neurons in the SNc

Every fourth section of the SN was stained for TH (approximately 10 sections per animal). We acquired the slide scans of each coronal brain section using a Zeiss Axio Observer.Z1 Inverted Microscope (Zeiss, Oberkochen, Germany) supported by a Yokogawa CSU-X1 unit (Yokogawa Life Science, Musashino-shi, Tokyo, Japan), enabling confocal imaging. It employed a 20×/0.8 NA objective lens for imaging. Seven Z-levels were imaged at 1 µm intervals and subsequently processed as maximum intensity projections using Zeiss Zen 3.1 software (Zeiss). An investigator blinded to the treatment groups manually quantified the number of TH-positive neurons in the SNc using the Zeiss Zen 3.1 software. The total number of TH-positive neurons per hemisphere was counted, and the sum in the treated hemisphere was expressed relative to that of the untreated hemisphere.

The SNr contains dopaminergic dendrites that belong to dopaminergic neurons in the SNc. We quantified the density of dopaminergic dendrites in the SNr using the above-mentioned slide scans. Image analysis was performed using the open-source software QuPath (version 0.5.1) [24]. The SNr was encircled in both hemispheres as an area of interest. To train the neural network-based pixel classifier of QuPath, TH-positive structures within the SNr were manually annotated in a subset of randomly selected sections. Next, the TH-positive area for each area of interest was determined utilizing the pixel classifier. The mean TH-positive area in the SNr per hemisphere was calculated for each mouse. The results from the treated hemisphere were normalized to the corresponding values from the non-treated hemisphere.

Dopaminergic neurons extend their axons into the striatum. To assess the integrity of dopaminergic axon terminals within the striatum, we acquired 10 to 15 high-magnification images (40×/0.95 objective) from two to three dorsal striatum-containing sections per mouse using a spinning disk confocal microscope. The imaging system consisted of a Zeiss Axio Observer.Z1 Inverted Microscope (Zeiss) equipped with a Yokogawa CSU-X1 unit (Yokogawa Life Science, Musashino-shi, Tokyo). Image analysis was performed in CellProfiler (version 4.2.4; Stirling et al., 2021 [25]). First, we used the “Enhance Neurites” function to define the borders of linear structures (e.g., axons) more clearly and applied a threshold to the image using the “robust background” algorithm. For each image, we calculated the ratio of positive pixels to the total image size. We calculated the mean value for each hemisphere of each animal. The results for each animal were then expressed as the ratio of the injected hemisphere to the contralateral hemisphere.

### 2.8. Evaluation of αSyn Pathology, Astrogliosis, and Microgliosis

Every fourth section spanning the entire SN (approx. 10 sections per animal) was stained for either H-αSyn in combination with P-αSyn and a nuclei counterstain, or for TH, Iba1and GFAP. Slide scans were acquired as described above. The SNc in both hemispheres was encircled as a region of interest in QuPath. The cell detection function was applied using default parameters, enabling the identification of nuclei based on Hoechst staining. To train the built-in object classifier algorithm for detecting cells positive for H-αSyn or P-αSyn, we manually annotated cells in a subset of slices. The entire SNc was then analyzed using the QuPath algorithm. The number of H-αSyn- and P-αSyn-positive cells per animal was summed for each hemisphere and then multiplied by four to account for the fact that every fourth section was analyzed. For the analysis of microgliosis and astrogliosis, we could not use nuclear staining as a basis for further analysis because all channels were used for the analysis of Iba1, GFAP, TH, and rAAV-EGFP. We therefore needed to modify our approach. The Iba1-positive area and the GFAP-positive area in the SNc were quantified by QuPath’s pixel classifier after training as outlined for αSyn. The relative Iba1-positive area and GFAP-positive area was quantified for each section. The values were summarized by calculating the mean for each hemisphere of each animal. Finally, for each animal, the mean value from the treated hemisphere was normalized to that of the corresponding non-treated hemisphere.

### 2.9. Analysis of p62-Positive, LAMP1-Positive, and RFP-EGFP-LC3-Positive Structures

In order to study the effect of PLEKHM1 overexpression on intracellular vesicles, we transduced RFP-EGFP-LC3 mice with rAAV-PLEKHM1. The RFP-EGFP-tagged LC3 enables pH-sensitive discrimination between autophagosomes and autolysosomes. EGFP fluorescence is quenched at acidic pH, resulting in autophagosomes appearing EGFP-positive and RFP-positive, while autolysosomes lose EGFP fluorescence and retain RFP fluorescence. Mature lysosomes are LAMP1-positive. We transduced six autophagy reporter mice and analyzed four sections per animal using the spinning disk confocal microscope described above, equipped with a 40×/0.95 NA objective. For each section, we acquired 15 to 20 images of the SNc in both hemispheres. Individual cells were cropped using Fiji (v2.16.0; [26]) and channels were split for further analysis. Cropped images depicting individual cells (*n* = 1239 cells) were processed using CellProfiler’s “enhance speckles” function and thresholded using the “robust background method”. The resulting binary images were processed with one opening step to minimize salt-and-pepper noise.

The area that represents lysosomes (LAMP1-positive, RFP-negative) was determined by the subtraction of binary images: RFP-positive pixels were subtracted from LAMP1-positive pixels, thus resulting in a binary image showing only those pixels that are LAMP1-positive and RFP-negative. Using the same approach, autolysosomes were defined by the subtraction of EGFP-positive pixels from the RFP-positive area. Autophagosomes were defined as EGFP and RFP-positive pixels. For each image, the number of positive pixels was divided by the total image area.

Using the same imaging, processing, and analysis workflow described above, the area covered by p62-positive structures was quantified in wild-type mice transduced with rAAV-EGFP or rAAV-PLEKHM1 (*n* = 1604 images depicting cells). In addition, autophagy reporter mice (RFP-EGFP-LC3) were analyzed (*n* = 1482). Signal intensity was also measured in cropped images depicting individual cells.

Colocalization analysis was performed using CellProfiler’s colocalization tool, employing default settings.

### 2.10. Statistical Analysis and Data Visualization

Data analysis and visualization were conducted in R (version 4.2.0) using RStudio (“Spotted Wakerobin” Release, version 2022.07.1+554) with the following packages: Tidyverse (version 2.0.0) for data wrangling and visualization, and Ggpubr (version 0.6.1) and FSA (version 0.10.0) for statistical testing [27,28,29]. For data that followed normal distribution, we used Welch’s two-sample *t*-test or a two-way ANOVA, followed by post hoc analyses with Tukey’s HSD or Dunn–Bonferroni tests. Pairwise comparisons that did not satisfy the aforementioned criteria were evaluated using the Wilcoxon rank sum exact test. For data involving more than two groups, the Kruskal–Wallis Chi-squared test was applied, followed by post hoc pairwise comparisons using Dunn’s Kruskal–Wallis multiple-comparison test with Benjamini–Hochberg adjustment. Correlations were calculated using Pearson’s product–moment correlation. Error bars represent the standard error of the mean. The *p*-values are listed in the figure legends. Graphical illustrations were created using Draw.io (v25.0.1., JGraph Ltd., Northampton, UK) and Microsoft PowerPoint 2021 (Microsoft Corporation, Redmond, WA, USA).

## 3. Results

### 3.1. αSyn Pathology Is Independent of PLEKHM1 Overexpression

We used stereotactic injections of rAAV under the neuronal synapsin1 promoter to express PLEKHM1 in the substantia nigra (SN; Figure 1A). Transgene expression was confirmed by immunohistochemical staining for the HA-tag of PLEKHM1 (Figure 1B), demonstrating robust expression within the targeted region. As a negative control, an rAAV-encoding EGFP was used, and EGFP expression was verified by fluorescence microscopy (Figure 1C). To induce αSyn pathology, we similarly injected rAAV expressing the A53T mutant form of αSyn under the same neuronal promoter. Human αSyn expression was detected via immunostaining and confirmed transgene expression in the cells of SN (Figure 1D,E). This approach allowed us to evaluate the effects of PLEKHM1 overexpression on αSyn pathology within the SN. Pathological intracellular αSyn aggregates are commonly hyperphosphorylated at serine 129 (P-αSyn; [30]). This post-translational modification is an established marker for αSyn pathology [31,32]. P-αSyn-positive cells were detected in both cohorts transduced with rAAV-αSyn (rAAV-αSyn + rAAV-PLEKHM1 and rAAV-αSyn + rAAV-PLEKHM1), confirming the occurrence of αSyn pathology (Figure 1D,E). Based on previous results, we hypothesized that the overexpression of PLEKHM1 alters autophagy and subsequently changes the clearance of aSyn aggregates. Nevertheless, the number of P-αSyn-positive cells was similar in animals transduced with PLEKHM1 (rAAV-αSyn + rAAV-PLEKHM1) as in animals transduced with EGFP (rAAV-αSyn + rAAV-EGFP) (Figure 1G). The number of cells transduced with αSyn was also similar, as evidenced by the number of H-αSyn-positive cells (Figure 1F).

### 3.2. PLEKHM1 Overexpression Exacerbates αSyn-Induced Degeneration of the Nigrostriatal Pathway

To histologically assess the dopaminergic system, we performed staining for tyrosine hydroxylase (TH). When the SN was transduced with EGFP, the number of TH-positive neurons in the SNc (red in Figure 2A) was unchanged compared to the uninjected hemisphere (Figure 2B), suggesting that viral transduction alone did not cause neurodegeneration. Transduction with PLEKHM1 (rAAV-EGFP + rAAV-PLEKHM1) or αSyn (rAAV-EGFP + rAAV-αSyn) reduced the number of TH-positive neurons in some animals, although this effect was not statistically significant at the group level (Figure 2B and Appendix A). Mice transduced with both αSyn and PLEKHM1 (rAAV-αSyn + rAAV-PLEKHM1) showed significantly fewer TH-positive neurons in the SN than the controls (rAAV-EGFP), or mice transduced with αSyn and EGFP (rAAV-αSyn + rAAV-EGFP) (Figure 2B). This observation can be explained by the hypothesis that PLEKHM1 potentiates the toxicity of αSyn.

The dendrites of dopaminergic neurons extend long apical dendrites into the SNr (orange in Figure 2A). We, therefore, measured the density of TH-positive dendrites in the SNr. In contrast to the analysis of neuron number, the expression of αSyn alone was sufficient to cause a significant reduction in the density of TH-positive dendrites in the SNr (Figure 2C). With the transduction of αSyn and PLEKHM1, the density of TH-positive dendrites in the SNr was also reduced as compared to the controls, but it was not significantly lower than with αSyn and EGFP (Figure 2C). We, therefore, interpret this finding as mainly representing αSyn toxicity.

The integrity of dopaminergic axon terminals was assessed by measuring the density of TH-positive structures in the striatum using images acquired at high magnification (Figure 2D). Regarding the neuron numbers in the SNc (Figure 2B), the co-expression of αSyn and PLEKHM1 reduced the density of TH-positive axon terminals in the striatum, whereas the individual expression of either αSyn or PLEKHM1 did not (Figure 2E).

The degeneration of dopaminergic somata, axons, and dendrites is closely correlated (Figure 2F,G). The loss of dendrites and cell somata is more pronounced than axonal degeneration, which may be explained by axonal sprouting compensating for striatal degeneration.

### 3.3. Increased Microglial Neuroinflammation in Mice Transduced with rAAV-αSyn + rAAV-PLEKHM1

Neuroinflammation is a key aspect of synucleinopathies. Microglial cells are the resident immune cells in the central nervous system and can convey protective and toxic effects [33,34,35,36]. In order to investigate microglia activation in our model, we stained for the established microglia marker ionized calcium-binding adapter molecule 1 (Iba1; Figure 3A).

The Iba1-positive area in the SNc was increased two-fold compared to the uninjected hemisphere in all groups (Figure 3B), suggesting a microglial response to viral transduction. In mice transduced with αSyn and PLEKHM1, the increase in Iba1-positive area was even greater than in control mice injected with the same number of viral particles (Figure 3B). Accordingly, we observed a negative correlation between the Iba1-positive area in the SNc and the density of TH-positive axon terminals in the striatum (Figure 3C). This correlation might be caused by the toxic effects of microglial activation, or by microglial activation downstream of neurodegeneration.

Activation of astroglia was quantified by staining for the intracellular intermediate filament protein ‘glial fibrillary acidic protein’ (GFAP; Figure 3D). The GFAP-positive area in the SNc was higher than in the uninjected hemisphere for all treatment groups; there was no significant difference in group-wise comparisons (Figure 3E). Nevertheless, we did observe a significant correlation between the GFAP-positive area in the SNc and the density of TH-positive dendrites in the SNr (Figure 3F).

### 3.4. PLEKHM1 Overexpression Impairs Autolysosome Maturation

To better understand the impact of PLEKHM1 overexpression on the autophagy-lysosomal system, we used RFP-EGFP-LC3 reporter mice (Figure 4A). These mice express the autophagosome marker, microtubule-associated protein 1A/1B-light chain 3 (LC3), fused to tandem RFP-EGFP fluorescence under the control of a CAG promoter [23]. The RFP-EGFP-tagged LC3 enables pH-sensitive discrimination between autophagosomes and autolysosomes. EGFP fluorescence is quenched at acidic pH, resulting in autophagosomes appearing EGFP-positive and RFP-positive, while autolysosomes lose EGFP fluorescence and retain RFP fluorescence (Figure 4B). We used pH-sensitive tandem fluorescence previously in vitro [37].

Neurons in the SN of RFP-EGFP-LC3 reporter mice were transduced by stereotactic injection of rAAV-PLEKHM1. In addition to the RFP-EGFP-LC3 fluorescence, we stained for the lysosomal marker LAMP1 and the nuclear marker DAPI (Figure 4C). Transduced cells in the SNc were compared to cells in the non-treated hemisphere using confocal microscopy. The area positive for RFP and LAMP1 in the SNc was reduced following rAAV-PLEKHM1 transduction, whereas the area positive for EGFP remained unchanged (Figure 4D).

We then examined these findings in more detail. Autophagosomes were defined as being EGFP-positive and RFP-positive. We did not observe a relevant difference between cells with and without rAAV-PLEKHM1 transduction (Figure 4E). Autolysosomes were defined as being RFP-positive and EGFP-negative. The area of autolysosomes was significantly decreased in cells with rAVV-PLEKHM1 transduction compared to the uninjected hemisphere (Figure 4F). Lysosomes were defined as RFP-negative and LAMP1-positive. The area of lysosomes was lower in cells with rAVV-PLEKHM1 transduction as compared to the uninjected hemisphere (Figure 4G). Finally, we noted an increase in the colocalization coefficient for LAMP1 with RFP (Figure 4H) and a decrease in the ratio of autolysosomes to autophagosomes (Figure 4I). In summary, the area corresponding to acidic vesicles (lysosomes and autolysosomes) was decreased, but the colocalization of LAMP1 with RFP was increased.

### 3.5. p62 Accumulation Reveals Impaired Autophagic Flux upon PLEKHM1 Overexpression

In the next step, we investigated the functional consequences of the observed changes in wild-type mice, as well as the autophagy reporter mice. To this end, we utilized p62 (also known as SQSTM1) as a marker (Figure 5A,B) [38]. p62 is a selective autophagy adaptor protein that serves as a surrogate marker for autophagic flux. It is continuously degraded under normal autophagic conditions and accumulates when autophagy is impaired.

p62 signal intensity showed a mild increase in rAAV-EGFP-transduced hemispheres compared to the untreated, contralateral hemisphere, indicating that viral transduction can modestly elevate baseline p62 levels (Figure 5C). Crucially, rAAV-PLEKHM1 transduction resulted in both genotypes in a pronounced, up to threefold increase in p62 intensity (Figure 5C,D). Next, we quantified the area per image in the SNc that was positive for p62 staining across both genotypes (Figure 5E,F). This area was comparable between untreated hemispheres and those injected with the control vector (rAAV-EGFP), suggesting that baseline p62 distribution remains largely unaffected by rAAV transduction alone. However, in both wild-type and transgenic autophagy reporter mice, rAAV-PLEKHM1 transduction led to a significant reduction in the p62-positive area. These findings support our visual observations of a shift in p62 staining from a diffuse cytoplasmic pattern to a markedly brighter and more punctate, irregular distribution upon PLEKHM1 overexpression.

Furthermore, we analyzed the colocalization of p62 with LAMP1 in wild-type mice, as well as the colocalization of p62 with LC3 in transgenic mice (Figure 5G–I). These analyses revealed an increase in p62-LAMP1 colocalization in wild-type mice transduced with rAAV-EGFP compared with the untreated hemisphere, which was markedly reduced following Plekhm1 overexpression. In autophagy reporter mice, p62 colocalization with autophagosomes (LC3EGFP+/RFP+) was significantly decreased, whereas colocalization with autolysosomes (LC3EGFP−/RFP+) was increased.

## 4. Discussion

In this study, we investigated the effect of PLEKHM1 overexpression in an αSyn-based mouse model of PD. The co-transduction of neurons with PLEKHM1 aggravated αSyn-induced neurodegeneration and neuroinflammation. Transduction with rAAV-PLEKHM1 also affected lysosomes and the maturation of autolysosomes, and impaired autophagic flux.

Transduction of neurons in the SN with rAAV-encoding αSyn-induced αSyn pathology, as evidenced by the presence of P-αSyn-positive cells. We examined the density of axon terminals and dendrites, as well as the number of dopaminergic neurons, to capture different stages of neurodegeneration and examine their spatial distribution. αSyn pathology was associated with a loss of dopaminergic dendrites in the SNr, a finding consistent with both our previous studies and those reported by other groups [21,22,39]. In contrast to earlier studies, we did not observe statistically significant degeneration of dopaminergic neuronal cell bodies in the SNc or a significant loss of axon terminals in the striatum. One possible explanation is that αSyn pathology causes early synaptic or dendritic dysfunction, preceding overt cell death and requiring additional stressors or prolonged exposure for neuronal loss to occur [40,41,42,43]. The deviation in the extent of toxicity observed with respect to our previous study [21] may also reflect technical factors such as viral vector quality, or biological factors like mouse strain susceptibility. On the other hand, a more pronounced baseline phenotype may have actually masked the effect of PLEKHM1 overexpression.

The overexpression of PLEKHM1, together with αSyn, induced the degeneration of dopaminergic neuron somata in the SNc and axon terminals in the striatum—an effect not observed with either PLEKHM1 or αSyn alone—indicating that αSyn-induced pathology alone is less potent in causing degeneration without a secondary insult. Interestingly, the increase in αSyn toxicity induced by PLEKHM1 overexpression was not accompanied by an increase in αSyn pathology, as measured by P-αSyn staining. This was unexpected, considering the established role of PLEKHM1 in lysosomal function [16] and the role of lysosomal degradation for αSyn pathology [44]. For instance, in HeLa cells, the siRNA-mediated depletion of PLEKHM1 slowed the removal of puromycin-induced p62-positive/ubiquitin-positive punctae, increasing aggregate count, size, and stability. In mouse embryonic fibroblasts from PLEKHM1-knockout mice (PLEKHM1^−^/^−^), the loss of PLEKHM1 resulted in decreased autolysosome formation under starvation conditions [16]. These discrepancies may be due to the differences between cultured cells and the mouse brain, since previous studies mostly include knockdown or knockout approaches in vitro, while we employed overexpression in vivo. Alternatively, aspects of αSyn pathology (e.g., oligomeric species) might mediate the toxic effects, which are not fully assessed using the readout of p-αSyn-positive cells. Crucially, signalling cascades such as the autophagy–lysosomal systems are tightly regulated, so both the knockdown and overexpression of PLEKHM1 might impair autophagy–lysosomal clearance.

To study the effect of elevated PLEKHM1 levels on the autophagy pathway in the mouse brain, we used RFP-EGFP-LC3 reporter mice, which allow for the identification of autophagosomes and autolysosomes. Lysosomes were identified by LAMP1 positivity, though LAMP1 can also be present on other endocytic compartments like multivesicular bodies and multilamellar bodies [45]. The overexpression of PLEKHM1 led to increased colocalization between LAMP1 and RFP, suggesting a blockade in the autophagy–lysosomal cycle at the autolysosome stage. This may impair lysosome reformation and contribute to lysosomal depletion, as reflected by the reduced LAMP1-positive area. Alternatively, the reduced LAMP1-positive area may indicate impaired lysosomal maturation. Additionally, the observed decrease in the autolysosome-to-autophagosome ratio implies that excessive PLEKHM1 disrupts autophagosome–lysosome fusion. Under normal conditions, PLEKHM1 coordinates with Rab7 and the HOPS complex to tether lysosomes to autophagosomes [16]. However, overexpression may disturb this stoichiometric balance, sequestering Rab7 or HOPS components into non-productive complexes. This could impair proper localization and function of the fusion machinery, leading to vesicle accumulation and ultimately impairing autophagic clearance. For instance, enlarged lysosomes were observed with constitutive overexpression of PLEKHM1 in U2OS cells and the majority of PLEKHM1-positive vesicles clustered in the perinuclear region and were less mobile [16].

To further understand the functional consequence of the observed changes, we used p62 as a surrogate marker for autophagic flux [38]. Upon PLEKHM1 overexpression, we observed p62 accumulation and a shift from a primarily cytoplasmic distribution to a brighter and irregular staining pattern. The increase in p62, evidenced by a rise in signal intensity, reflects impaired clearance and subsequent accumulation, as typically observed upon autophagy inhibition [46]. This effect was consistent across both genotypes, although rAAV-EGFP exhibited a slight increase in p62 intensity, but not in area. Colocalization analysis of p62 with LAMP1 and autophagosomes (LC3EGFP+/RFP+), and autolysosomes (LC3EGFP−/RFP+) revealed a reduction in overlap with both LAMP1 and autophagosomal, but an increase in colocalization with autolysosomal markers. The observed reduction in p62-LAMP1 colocalization may reflect decreased delivery of p62 to lysosomes, potentially compromised lysosomal biogenesis (note the reduced LAMP1-positive area) or the activation of alternative degradation pathways as a compensatory mechanism. However, p62 that reaches autolysosomes appears to accumulate, suggesting it becomes trapped and indicating a defect in autophagic flux. Our finding is also consistent with the compromised clearance of lysosomal cargo, including epidermal growth factor receptor with PLEKHM1 overexpression [47]. In fact, altered growth factor trafficking could underlie the observed increase in neurodegeneration without effects on αSyn pathology.

rAAV vectors can activate microglia, as confirmed by our previous study [21]. To control for this, we included an rAAV-EGFP group. Microglial activation was similar in rAAV-EGFP and rAAV-PLEKHM1 groups, showing PLEKHM1 does not increase immune activation. Interestingly, the increased microglial activation observed in mice co-expressing αSyn and PLEKHM1 might contribute to dopaminergic neuron degeneration. However, this heightened response could also be a secondary consequence of the increased neuronal loss. The synapsin1 promoter used in the rAAV for αSyn and PLEKHM1 primarily drives expression in neurons, suggesting that microglia activation is not a direct effect of αSyn or PLEKHM1 expression in microglia [48].

Our finding that PLEKHM1 potentiates the degeneration of dopaminergic neurons is, in fact, consistent with genetic and epigenetic studies. The increased expression of PLEKHM1 is a risk factor for PD [11], whereas methylation of the PLEKHM1 locus was negatively associated with PD risk [12]. At the same time, however, the reduced expression of PLEKHM1 has also been linked to disease, consistent with the cell culture data summarized above. Dysregulated transcription factor binding due to a SNP in the promoter region of PLEKHM1 in human neurons has been associated with an increased risk for PSP [13], and SNPs that increase the risk of PD were associated with reduced PLEKHM1 mRNA in the cerebellum [49].

Finally, the link between PLEKHM1 and exacerbated neurodegeneration needs to be explored in further studies. One could hypothesize that PLEKHM1 overexpression serves as a first hit, rendering neurons more vulnerable and reducing their resilience to αSyn pathology. For instance, autophagy inhibition leads to accumulation of the adaptor protein p62, which activates pro-inflammatory pathways such as NF-κB and Nrf2 [50,51]. Furthermore, lysosomal dysfunction as observed with PLEKHM1 overexpression can lead to swelling and rupture, releasing cathepsins and other hydrolases into the cytoplasm and triggering cell death cascades [36,52]. Elevated PLEKHM1 may impair mitophagy, causing the accumulation of ROS-producing mitochondria and oxidative stress, similar to nigrostriatal degeneration due to PINK1/Parkin deficiency [53]. Lastly, increased microglial activation, as observed in the αSyn + PLEKHM1 condition, may contribute to the cell loss and has even been shown to induce dopaminergic neuron loss independently of αSyn [54].

This study has several limitations. First, αSyn pathology could be more comprehensively characterized, for instance, by examining its spatial distribution (e.g., in the striatum) or by identifying the αSyn species involved. Second, autophagy-related comparisons in reporter mice were based on the differences between injected and non-injected hemispheres, without the use of a control injection. Nevertheless, experiments in wild-type mice, including measurements of autophagic flux, were appropriately controlled. The co-expression of αSyn with PLEKHM1 in this background could provide additional mechanistic insight. Third, sample sizes were relatively small, limiting statistical power. Finally, while the findings support a mechanistic hypothesis, the exact molecular link between PLEKHM1 overexpression and neurodegeneration remains to be fully defined.

Taken together, our findings suggest that elevated PLEKHM1 levels reduce autophagic flux in wild-type as well as autophagy reporter mice, and exacerbate neurodegeneration in αSyn-overexpressing mice. Combined with existing genetic evidence, this supports the idea that both up- and downregulation of PLEKHM1 impair autophagic clearance, increasing neuronal vulnerability to secondary hits such as αSyn pathology.

## 5. Conclusions

Altered PLEKHM1 expression is linked to neurodegenerative diseases;PLEKHM1 was studied in autophagy reporter mice and a Parkinson’s disease mouse model;PLEKHM1 overexpression worsens α-synuclein-induced neurodegeneration and inflammation;Elevated PLEKHM1 compromises autophagic flux.

## Figures and Tables

**Figure 1 cells-14-01340-f001:**
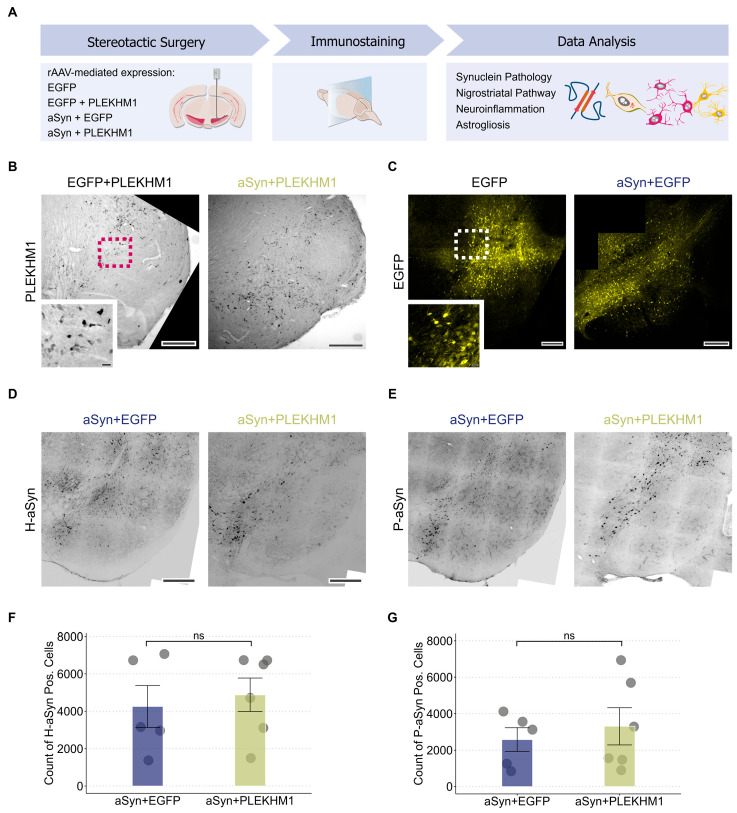
αSyn pathology is independent of PLEKHM1 overexpression. (**A**) Wild-type mice received unilateral stereotactic injections of recombinant adeno-associated virus into the substantia nigra pars compacta. Neuronal transduction was achieved using a 1:1 mixture of the indicated viral vectors. Eight weeks post-injection, synuclein pathology, nigrostriatal pathway integrity, neuroinflammation, and astrogliosis were evaluated via immunohistochemistry and immunofluorescence. (**B**) Representative brightfield image of immunohistochemical staining for HA-tagged PLEKHM1 in the SN confirming transgene expression after transduction with rAAV-PLEKHM1. Scale bar = 300 µm. Inset indicated by the pink dotted square in the non-magnified image. Scale bar = 10 µm. (**C**) Representative slide scan showing EGFP expression in neurons of the SN following transduction with rAAV-EGFP. Scale bar = 300 µm. Inset indicated by the white dotted square in the non-magnified image. Scale bar = 20 µm. (**D**,**E**) Representative slide scans depicting the SN in animals transduced with rAAV-αSyn + rAAV-EGFP and rAAV-αSyn + rAAV-PLEKHM1. H-αSyn-positive (**D**) and P-αSyn-positive cells (**E**) were visualized by immunofluorescence. Scale bar = 200 μm. (**F**,**G**) Quantification of H-αSyn-positive cells (**F**) and P-αSyn-positive cells (**G**) in the SNc. Dots represent individual animals (*n* = 5 in rAAV-αSyn + rAAV-EGFP, *n* = 6 in rAAV-αSyn + rAAV-PLEKHM1). Non-significant results are annotated with “ns”.

**Figure 2 cells-14-01340-f002:**
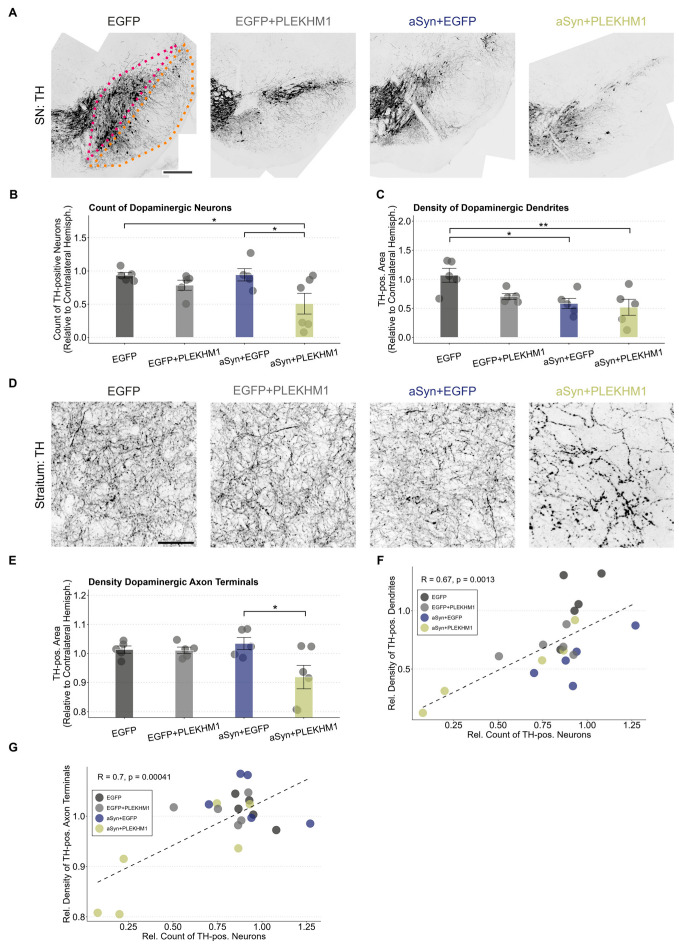
PLEKHM1 overexpression exacerbates the αSyn-induced degeneration of the nigrostriatal pathway. (**A**) Representative images of the SN from the treated hemisphere. Dopaminergic neurons in the midbrain were visualized using immunofluorescence staining for tyrosine hydroxylase. Cohorts are indicated above each image. The SNc is highlighted in pink and the SNr is outlined in orange. Scale bar: 300 μm. (**B**) Count of TH-positive neurons in the SNc. Each animal’s count of TH-positive neurons in the transduced hemisphere is shown as a ratio to the contralateral hemisphere. Dots represent individual animals (*n* = 5 in all cohorts, except *n* = 6 in mice transduced with rAAV-αSyn + rAAV-PLEKHM1). Comparison by a two-way ANOVA. Independent variable 1: rAAV-EGFP or rAAV-αSyn. Independent variable 2: rAAV-EGFP or rAAV-PLEKHM1. Variable 2 explains the significant variation (F-Value = 7.572, *p*-value = 0.0136). The interaction between these variables was not significant. Tukey’s HSD post hoc analysis revealed significant pairwise differences between rAAV-αSyn + rAAV-EGFP and rAAV-αSyn + rAAV-PLEKHM1 (*p*-value = 0.045), as well as rAAV-EGFP and rAAV-αSyn + rAAV-PLEKHM1 (*p*-value = 0.049). Asterisks indicate significance levels (* *p* < 0.05). (**C**) Density of TH-positive area in the SNr normalized to the contralateral hemisphere. Dendrites of dopaminergic neurons are located in the SNr. Dots represent individual animals (*n* = 5 in all cohorts, except *n* = 6 in mice transduced with rAAV-αSyn + rAAV-PLEKHM1). Comparison by a two-way ANOVA. Independent variable 1: rAAV-EGFP or rAAV-αSyn. Independent variable 2: rAAV-EGFP or rAAV-PLEKHM1. Variable 1 explains the significant variation (F-value = 10.363, *p*-value = 0.005). The interaction between these variables was not significant. Tukey’s HSD post-hoc analysis revealed significant pairwise differences between rAAV-EGFP and rAAV-αSyn + rAAV-EGFP (*p*-value = 0.021) as well as rAAV-EGFP and rAAV-αSyn + rAAV-PLEKHM1 (*p*-value = 0.009). Asterisks indicate significance levels (* *p* < 0.05, ** *p* < 0.01). (**D**) Representative images of dopaminergic axons and axon terminals in the striatum labeled by TH. Cohorts as indicated above each image. Scale bar: 30 μm. (**E**) Density of TH-positive area in the striatum, normalized to the contralateral hemisphere. Comparison by a two-way ANOVA. Independent variable 1: rAAV-EGFP or rAAV-αSyn. Independent variable 2: rAAV-EGFP or rAAV-PLEKHM1. Variable 2 explains the significant variation (F-value = 5.394, *p*-value = 0.0329). The interaction between these variables was significant (F-value = 4.511, *p*-value = 0.0487). Tukey’s HSD post hoc analysis revealed significant pairwise differences between rAAV-αSyn + rAAV-EGFP and rAAV-αSyn + rAAV-PLEKHM1 (*p*-value = 0.0273). Dots represent individual animals. Asterisks indicate significance levels (* *p* < 0.05). (**F**) Correlation between the count of dopaminergic neurons and the density of dopaminergic dendrites. Each dot represents an individual animal. Pearson’s product–moment correlation with R = 0.67 (*p*-value = 0.0013). Dotted line represents the corresponding generalized linear mixed model. (**G**) Correlation of the count of dopaminergic neurons and dopaminergic axon terminal density. Each dot represents an individual animal. Pearson’s product–moment correlation with R = 0.7 (*p*-value = 0.00041). Dotted line represents the corresponding generalized linear mixed model.

**Figure 3 cells-14-01340-f003:**
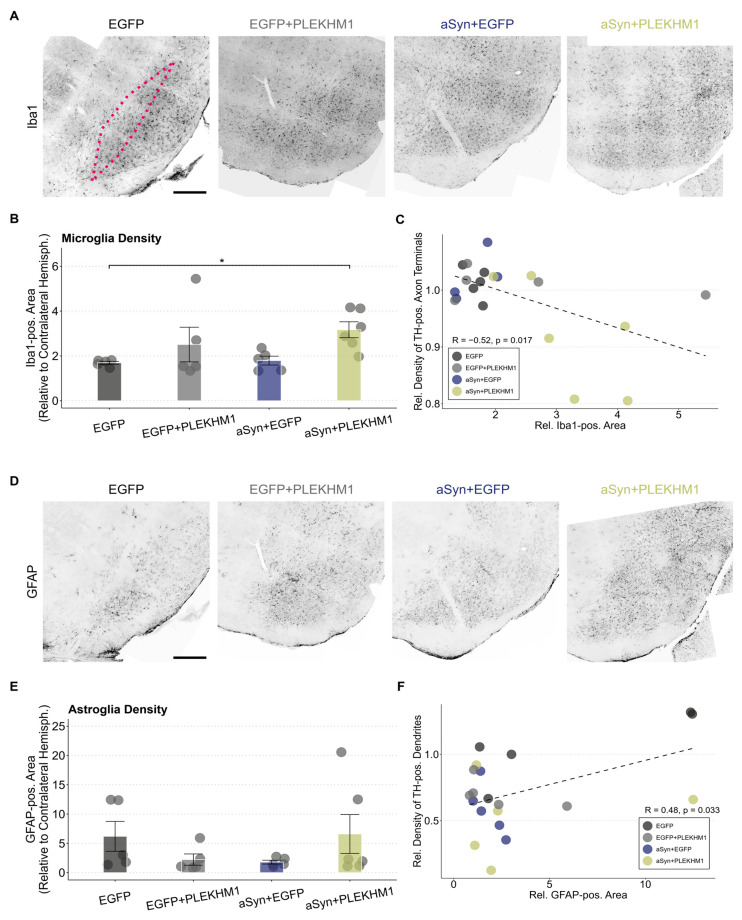
Increased microglial neuroinflammation in mice transduced with rAAV-αSyn + rAAV-PLEKHM1. (**A**,**D**) Representative images of Iba1-positive microglia (**A**) and GFAP-positive astrocytes (**D**) in the SN across all treatments. The SNc is highlighted by the pink dotted line. Cohorts are annotated above each image. Scale bar = 300 μm. (**B**) Iba1-positive area in the SNc as a ratio to the contralateral hemisphere. Dots represent individual animals (*n* = 5 in all cohorts, except *n* = 6 in mice transduced with rAAV-αSyn + rAAV-PLEKHM1). Comparison by a two-way ANOVA. Independent variable 1: rAAV-EGFP or rAAV-αSyn. Independent variable 2: rAAV-EGFP or rAAV-PLEKHM1. Variable 2 explains for a significant variation (F-value = 6.592, *p*-value = 0.02). Post hoc analysis by the Dunn test with Bonferroni correction for multiple testing suggests a significant pairwise difference between rAAV-EGFP and rAAV-αSyn + rAAV-PLEKHM1 (*p*-value = 0.042). Asterisks indicate significance levels (* *p* < 0.05). (**C**) Correlation of the Iba1-positive area and density of dopaminergic axon terminals. Each dot represents an individual animal. Pearson’s product–moment correlation with R = −0.52 (*p*-value = 0.017). Dotted line represents the corresponding generalized linear mixed model. (**E**) GFAP-positive area in the SNc relative to the contralateral hemisphere. No significant pairwise differences were found in statistical testing. Dots represent individual animals. (**F**) Correlation of the GFAP-positive area and the density of dopaminergic dendrites. Each dot represents an individual animal. Pearson’s product–moment correlation with R = 0.48 (*p*-value = 0.033). Dotted line represents the corresponding generalized linear mixed model.

**Figure 4 cells-14-01340-f004:**
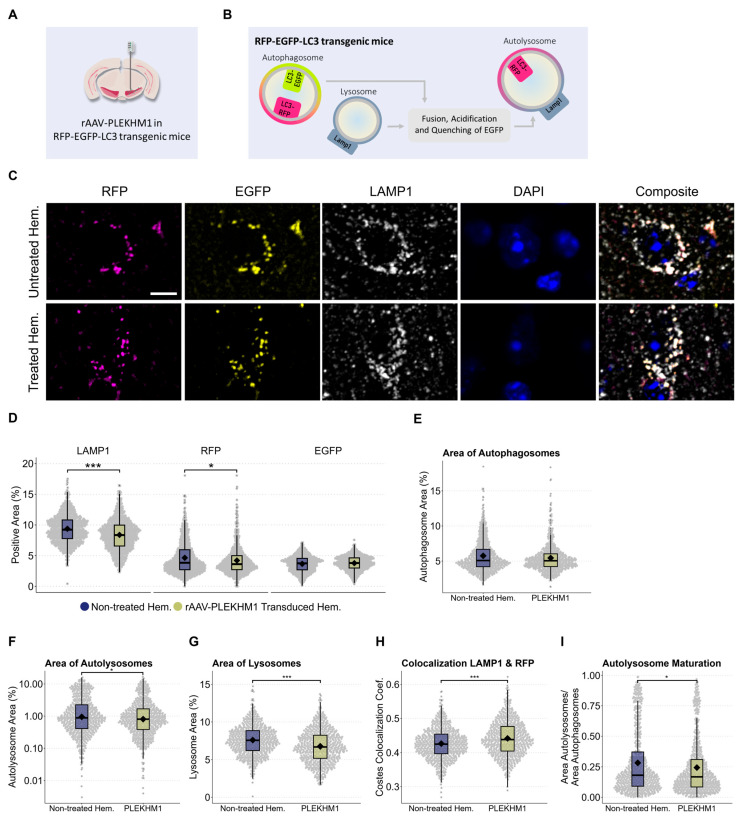
PLEKHM1 overexpression compromises lysosomes and impairs autolysosome maturation, as observed in autophagy reporter mice. (**A**) Illustration of the experimental design. Six RFP-EGFP-LC3 transgenic mice received unilateral stereotactic injections of rAAV-PLEKHM1 into the SNc and were analyzed after eight weeks. (**B**) These autophagy reporter mice express LC3 fused to both RFP and EGFP. The RFP-EGFP-tagged LC3 enables the pH-sensitive discrimination between autophagosomes and autolysosomes. EGFP fluorescence is quenched at acidic pH, resulting in autophagosomes appearing EGFP-positive and RFP-positive, while autolysosomes lose EGFP fluorescence and retain RFP fluorescence. (**C**) Representative images show neurons in the SNc from both hemispheres. Autophagosomes are identified by LC3 tagged with EGFP and RFP (EGFP^+^/RFP^+^), autolysosomes are marked by RFP-only LC3 signal (EGFP^−^/RFP^+^), and lysosomes are visualized using immunofluorescence staining for LAMP1 (LAMP1^+^/RFP^−^). Scale bar = 10 µm. (**D**) Analysis of the signal-positive area for each channel. The total LAMP1-positive area decreased significantly using the Wilcoxon rank-sum test (*p*-value = 5.7 × 10^−12^), as well as the total RFP-positive area (*p*-value = 0.021). (**E**) The signal-positive area representing autophagosomes (LC3^EGFP+/RFP+^) was evaluated and statistical analysis indicated no significant difference. (**F**) Evaluation of the signal-positive area representing autolysosomes (LC3^EGFP−/RFP+^) was performed. A statistically significant difference was detected using the Wilcoxon rank-sum test (*p* = 0.039). (**G**) Lysosomes were identified as LAMP1-positive structures that did not colocalize with LC3-RFP, which marks autolysosomes or autophagosomes. The Wilcoxon rank-sum test revealed a significant difference in the LAMP1-positive area (*p*-value = 2.2 × 10^−11^). (**H**) Colocalization analysis using the Costes method revealed a significant reduction in the colocalization of LAMP1 with RFP. This was statistically confirmed using the Wilcoxon rank-sum test (*p*-value = 3.7 × 10^−7^). (**I**) A statistically significant difference in the ratio of autolysosomes to autophagosomes was observed, as confirmed by the Wilcoxon rank-sum test (*p*-value = 0.031). Asterisks indicate significance levels (* *p* < 0.05, *** *p* < 0.001).

**Figure 5 cells-14-01340-f005:**
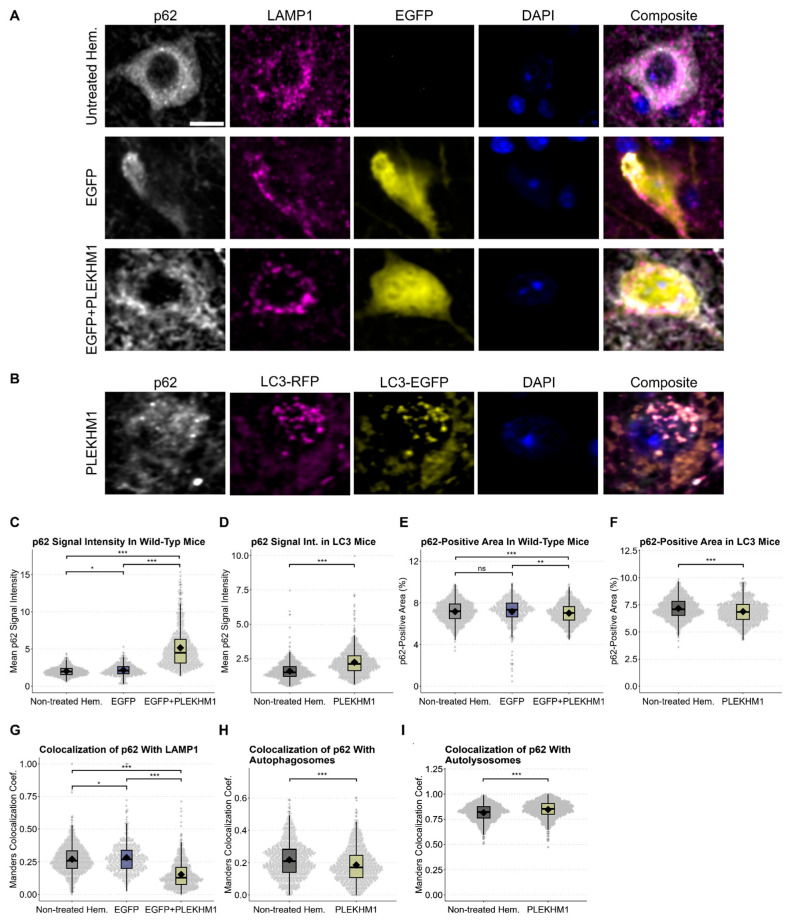
p62 accumulation reveals impaired autophagic flux upon PLEKHM1 overexpression. (**A**,**B**) Representative images depict cells in the SNc of wild-type mice, showing untreated hemispheres and those transduced with rAAV-EGFP or rAAV-EGFP + rAAV-PLEKHM1 in panel (**A**). Images from RFP-EGFP-LC3 mice transduced with rAAV-PLEKHM1 are shown in panel (**B**). p62-positive cargo, LAMP1-positive lysosomes, autophagosomes (LC3EGFP+/RFP+), and autolysosomes (LC3EGFP−/RFP+) were visualized. Scale bar = 10 µm. (**C**,**D**) Analysis of p62 signal intensity across conditions and genotypes demonstrated a significant increase in the rAAV-PLEKHM1-transduced mice compared to both the non-treated hemisphere and rAAV-EGFP controls. *n* = 1604 images depicting cells in wild-type mice and *n* = 1482 in transgenic mice. In wild types, statistical analysis revealed significant differences as determined by the Kruskal–Wallis Chi-squared test (*p* = 2.2 × 10^−16^). Post hoc pairwise comparisons were conducted using Dunn’s Kruskal–Wallis multiple-comparison test, with *p*-values adjusted using the Benjamini–Hochberg method. rAAV-EGFP vs. non-treated hem., *p* = 0.011; rAAV-EGFP vs. rAAV-EGFP + rAAV-PLEKHM1, *p* < 1 × 10^−96^; and non-treated hem. vs. rAAV-EGFP + rAAV-PLEKHM1, *p* < 1 × 10^−160^. In autophagy reporter mice, the Wilcoxon rank-sum test revealed a statistically significant increase in the p62 signal intensity between groups (*p* < 2.2 × 10^−16^). (**E**,**F**) Quantification of p62-positive area across treatments and genotypes revealed a significant reduction in total p62-positive area in hemispheres transduced with rAAV-EGFP + rAAV-PLEKHM1, whereas controls (non-treated hemisphere and rAAV-EGFP) showed no significant differences. In wild-type mice, statistical analysis using the Kruskal–Wallis Chi-squared test revealed significant differences among the groups (*p* = 9.816 × 10^−5^). Group-wise comparisons were subsequently performed using Dunn’s Kruskal–Wallis multiple comparison test, with *p*-values adjusted using the Benjamini–Hochberg method. rAAV-EGFP vs. non-treated hem., *p* = 0.0641; rAAV-EGFP vs. rAAV-EGFP + rAAV-PLEKHM1, *p* = 8.15 × 10^−5^; and non-treated hem. vs. rAAV-PLEKHM1, *p* = 7.97 × 10^−3^. In autophagy reporter mice, a Wilcoxon rank-sum test revealed a statistically significant decrease in the p62-positive area between the groups (*p* = 2.3 × 10^−8^). (**G**) Manders colocalization coefficient was increased p62-LAMP1 colocalization in rAAV-EGFP control hemispheres compared with the untreated hemisphere, whereas rAAV-PLEKHM1-transduced hemispheres showed a reduction. This effect was confirmed as determined by the Kruskal–Wallis Chi-squared test (*p* < 2.2 × 10^−16^). Post hoc pairwise comparisons using Dunn’s Kruskal–Wallis multiple comparison test with Benjamini-Hochberg adjustment yielded the following results: rAAV-EGFP vs. non-treated hem., *p* = 0.011; rAAV-EGFP vs. rAAV-EGFP + rAAV-PLEKHM1, *p* < 3 × 10^−97^; and non-treated hem. vs. rAAV-EGFP + rAAV-PLEKHM1, *p* < 3 × 10^−161^. (**H**,**I**) The Manders colocalization coefficient was significantly decreased for the overlap between p62 and autophagosomes (LC3EGFP+/RFP+; Wilcoxon rank-sum test, *p* < 1.1 × 10^−9^) and increased in p62 colocalization with autolysosomes (LC3EGFP−/RFP+; Wilcoxon rank-sum test, *p* < 1.1 × 10^−13^). Asterisks indicate significance levels (* *p* < 0.05, ** *p* < 0.01, *** *p* < 0.001).

## Data Availability

The data presented in this study will be made available on request.

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
