# Peer review of "PLEKHM1 Overexpression Impairs Autophagy and Exacerbates Neurodegeneration in rAAV-α-Synuclein Mice"

_cells, 2025, doi:10.3390/cells14171340_

Round 1
Reviewer 1 Report
Comments and Suggestions for Authors
The manuscript by Hofs et at. investigates the role of PLEKHM1 expression in autophagy and aSyn pathology in vivo mice. Given the known role of PLEKHM1 in mediating fusion events of autophagosomes with lysosomes, the authors hypothesised that PLEKHM1 overexpression might increase the autophagic flux and reduce aSyn pathology. The paper is well written, with a detailed methods section.
However, this reviewer has some comments and concerns:
- In figure 1, PLEKHM1 overexpression does not reduce aSyn pathology. As both aSyn and PLEKHM1 overexpression are achieved by the injection of two separate AAVs in the Substantia Nigra, the levels of pSer129 should be measured in the HA-PLEKHM1 positive neurons by co-staining of HA and pSer129 (immunofluorescence).
Also, the authors should consider to stain for anti-aggregated aSyn antibodies or perform western blot in the tissue and detect any insoluble aSyn aggregates. - All analysis were performed at eight weeks post injection. Have the authors considered earlier and later time points?
- Page 8, line 308: based on the significant reduction of TH+ neuron density in the SN, the authors introduce the hypothesis that PLEKHM1 potentiates the toxicity of aSyn, which is the opposite of the statement in the introduction. This should be treated carefully as it creates confusion in the general purpose of this study. In the introduction, the authors should make general claims about their hypothesis.
- Have the authors investigated aSyn pathology in the striatum to correlate their results on dopaminergic axonal degeneration induced by aSyn + PLEKHM1 overexpression.
- The title of paragraph 3.3 is misleading as the quantification of Iba1 does not show significant increased microglia neuroinflammation with PLEKHM1 (no significant difference in the comparison EGFP vs EGFP+PLEKHM1, nor aSyn vs aSyn+PLEKHM1). Also, a representative image of the non-injected hemisphere should be shown to justify the 2-fold increase in Iba1 due to AAVs injection per se. Same for GFAP staining.
- In the mice expressing the RFP-EGFP-LC3 construct, the authors should better clarify that autophagosomes are both RFP and EGFP positive (hence yellow), whilst acidic autolysosomes should be only RFP positive. On the same line, it is not clear whether figure 4D show the quantification of RFP+ only vs RFP+EGFP+ structures.
- When quantifying RFP-EGFP-LC3 reporter, the authors should include the number of puncta as parameter, rather than area.
- Why are lysosomes defined as RFP-negative and LAMP1-positive? LAMP1 has a wider distribution than lysosomes, i.e., multi-vesicular bodies, late endosomes, multi-lamellar bodies… (see Cheng et al., JCB 2018).
- In the figure 4, the analysis is done comparing injected hemisphere vs non injected only. A control for the injection per se (i.e. EGFP, as previous experiments) should be included.
- The statistical analysis in Figure 4 should be done on the mean value obtained from each animal rather than each cells analysis, as the difference between the means is not so evident and it might derive from the power of the test due to the high number of data points.
- In the discussion, the authors stated that their aSyn overexpressing model does not recapitulate aSyn-induced degeneration as previously described. This undermines the relevance of their study and any potential role by PLEKHM1 on aSyn pathology. The authors should clarify this as the significance of their study is not convincing in the current version of the manuscript, especially if their argument is that “discrepancies [with previous studies] can potentially be explained by differences between cultured cells and mouse brains.”” (page 14, lines 473-474)
Reviewer 2 Report
Comments and Suggestions for Authors
Höfs et al. investigate the impact of PLEKHM1 overexpression on α-synuclein–induced neurodegeneration and autophagic flux in mouse models. Using rAAV to co-express PLEKHM1 and A53T-αSyn in the substantia nigra, they show that PLEKHM1 exacerbates dopaminergic neuron loss, dendritic and axonal degeneration, and microglial activation, without altering the overall burden of phosphorylated α-synuclein. In transgenic RFP-EGFP-LC3 reporter mice, PLEKHM1 overexpression reduces autolysosome and lysosome abundance while increasing LAMP1–RFP colocalization, suggesting impaired autolysosome maturation. These findings imply that dysregulated PLEKHM1 levels may aggravate synucleinopathy by disrupting autophagic clearance. There are many critical issues that should be addressed, and numerous uncertainties remain regarding the interpretation of the results.
Specific issues:
1. Please verify the baseline expression level of PLEKHM1 in your target neuronal population. In addition, quantitatively report how much PLEKHM1 is upregulated in your overexpression system.
2. In Fig. 2A, the Z-positions of the axial sections appear to differ between groups. Please properly refine these alignments and re-examine the data for consistency.
3. In Fig. 2C you report a significant reduction in TH-positive dendritic density despite no accompanying loss in TH-positive neuron number. It is unclear why these two readouts diverge and what biological process underlies this. Please clarify the mechanistic rationale, for example by discussing whether α-synuclein overexpression triggers sublethal dendrite retraction or synaptic dysfunction before neuronal loss. You should also justify your choice of endpoints.
4. The use of the RFP–EGFP–LC3 reporter to assess autophagy is not sufficiently clarified. I recommend that the authors:
・Quantify and report the ratio of RFP⁺/GFP⁺ (autophagosomes) to RFP⁺/GFP⁻ (autolysosomes) puncta under each experimental condition, including representative micrographs and statistical analysis.
・Explain how shifts in this ratio reveal PLEKHM1’s function in autophagy flux, for example by describing whether PLEKHM1 affects fusion between autophagosomes and lysosomes or alters degradation efficiency.
・If this reporter mice alone cannot clearly establish the function of PLEKHM1 in autophagy pathway in this context, please provide additional experiments, such as LC3-II and p62 immunoblotting under both basal conditions and with lysosomal inhibition.
5. Please either experimentally investigate or, at a minimum, discuss in the Discussion section the mechanistic link between the observed neurodegeneration phenotype and PLEKHM1-mediated suppression of autophagy.
6. To exclude the possibility that unintended immune responses is driving your neurodegeneration readouts, please validate that neither rAAV transduction nor PLEKHM1 overexpression affects the immune status.
7. The figure numbering and the order of citations in the text are currently inconsistent. Please renumber and reorder all figures so that they follow the exact sequence.
Comments on the Quality of English LanguageThe overall readability would be greatly improved by professional language editing. I recommend that the authors engage a native‐level English speaker or a scientific copyeditor to eliminate minor grammatical errors and tighten phrasing for greater precision.
e.g.)
Line 40-41: As a result, promoting autolysosome formation to boost the clearance of toxic ”,” misfolded αSyn is considered a promising therapeutic strategy.
Line 50: Conversely, decreased PLEKHM1 expression due “to” promoter 50 Methylation…..
Reviewer 3 Report
Comments and Suggestions for Authors
The study explores the impact of overexpressing PLEKHM1, a protein involved in regulating the autophagy-lysosomal pathway, on αSyn-induced neurodegeneration using mouse models. The results indicate that PLEKHM1 overexpression exacerbates dopaminergic neuron degeneration, activates microglia, and impairs autolysosome maturation, which hinders the clearance of αSyn aggregates. Specifically, in the αSyn-PLEKHM1 co-expression model, significant degeneration of dopaminergic neurons and axon terminals in the substantia nigra and striatum was observed, yet no increase in αSyn pathology was detected. These findings suggest that while PLEKHM1 may influence αSyn-induced neurodegeneration, its role in autophagic clearance remains unclear. The results align with genetic studies linking PLEKHM1 expression to neurodegenerative diseases such as Parkinson’s disease. However, the study design presents certain limitations. Notably, the lack of significant effects on αSyn pathology despite clear neurodegeneration indicates that other factors, such as viral vector integrity or strain-specific variations, may have influenced the outcomes. Additional comments follow:
- Introduction: The authors cite GWAS findings linking PLEKHM1 SNPs to the risk of Alzheimer’s disease and Parkinson’s disease, while also mentioning a negative association between PLEKHM1 promoter methylation and Parkinson’s disease risk. However, the manuscript does not address the apparent contradiction in these genetic regulatory mechanisms or how they influence PLEKHM1 expression. Moreover, the biological significance of "increased expression" versus "decreased expression" of PLEKHM1 in different disease contexts is not clearly explained. It is recommended that the authors expand this discussion to clarify how these genetic factors interact to regulate PLEKHM1 expression and their implications for neurodegenerative diseases. This would offer a more comprehensive understanding of PLEKHM1’s complex role in disease pathogenesis.
- Methods: In selecting mutation sites, the authors substituted alanine with proline at positions 30, 56, and 76 (A30P/A56P/A76P) of hαSyn. However, the manuscript lacks references supporting the synergistic effect of these mutations on αSyn aggregation characteristics. Furthermore, the rationale for choosing these specific mutation sites over well-established pathogenic mutations, such as A53T or E46K, is not provided. It is recommended that the authors justify their choice of these mutations and include relevant literature to support their potential synergistic effects on αSyn aggregation.
- Figure 1: Although αSyn pathology is quantitatively analyzed (Figures 1F/G), it remains unclear whether this analysis accurately represents typical αSyn aggregation seen in PD. Additionally, the manuscript does not specify whether the cells analyzed are exclusively neurons or if glial cells were included. Since pathology characterization is a critical aspect of the study, the authors are encouraged to provide a more detailed explanation of the nature of the observed αSyn aggregation, clarify whether the analysis targets neuronal cells exclusively, and discuss the possible inclusion of glial cells in the pathology assessment.
- Figure 2B: The slight (non-significant) decreases in TH-positive neurons in the αSyn+EGFP group compared to the EGFP group contradicts the well-established literature indicating that α-synuclein overexpression causes significant dopaminergic neuron loss. Moreover, the figure does not assess α-synuclein protein aggregation or phosphorylation status, making it difficult to confirm whether the neuronal loss is directly linked to α-synuclein pathology. It is recommended that the authors provide a more detailed analysis, including assessment of α-synuclein aggregation and phosphorylation, to establish a clearer connection between αSyn pathology and dopaminergic neuron loss.
- Figure 2B: While significant loss of dopaminergic cell bodies in the αSyn+PLEKHM1 group (approximately 50% reduction in TH-positive neurons) is shown, Figure 2E only shows a slight decrease (approximately 10%) in axonal terminal density. This disproportionate pattern of degeneration does not align with the "dying-back" pathology typically observed in PD. Literature indicates that α-synuclein primarily affects presynaptic terminals, contradicting the observed results. The authors should address this inconsistency and explore the potential reasons for this divergent pattern of neuronal loss and its implications.
- Figure 3: The positive correlation between TH density and GFAP area (R = 0.48, p = 0.033) suggests that astrocyte activation is negatively associated with neuronal loss. This finding contradicts the established view that neurodegenerative damage induces astrocyte proliferation as a protective response. Additionally, Figure 3B shows that Iba1 density in the EGFP group (control group) is significantly higher than physiological levels, suggesting that the viral injection itself may have triggered microglial activation.
- Figure 4: The contrast of the microscope images in Figure 4C is low, and the boundaries between RFP and EGFP signals are unclear. Furthermore, the results do not include an autophagy analysis for the αSyn treatment group, complicating the assessment of whether PLEKHM1 affects the autophagic dysfunction induced by αSyn. In addition, Figure 4G shows a reduction in the LAMP1-positive lysosome area, suggesting that PLEKHM1 overexpression may interfere with lysosome biogenesis or maturation, rather than solely affecting the fusion process. This observation raises the possibility that PLEKHM1 overexpression disrupts multiple stages of the autophagy-lysosomal pathway.
- Discussion: The discussion lacks sufficient mechanistic depth in linking autophagic dysfunction to neurodegeneration. While PLEKHM1 overexpression impairs autophagic flux (Figure 4), the manuscript does not explain how this leads to the exacerbation of αSyn toxicity (Figure 2). The authors should discuss whether autophagy inhibition induces abnormal αSyn aggregation, particularly focusing on αSyn oligomerization and phosphorylation. Additionally, a more detailed exploration of how lysosomal dysfunction affects neuronal homeostasis, including calcium dysregulation or mitochondrial abnormalities, is needed to establish a clearer connection between autophagic impairment and neuronal degeneration.
Reviewer 4 Report
Comments and Suggestions for Authors
The study addresses a biologically meaningful and translationally relevant question on how PLEKHM1 overexpression impacts autophagy and α-synuclein-induced neurodegeneration, a central mechanism in Parkinson's disease (PD).
Although it is interesting, there are some major concerns need to be addressed by authors:
1, The central paradox, PLEKHM1 impairs autophagy but does not increase α-syn aggregates, is acknowledged but not well resolved. While compensatory mechanisms are suggested by authors, no supporting experiments (e.g., biochemical α-syn clearance, turnover assays) are included to verify this. The authors should add western blot or dot blot data for total αSyn or insoluble fractions to further illustrate the event and support conclusions beyond P-αSyn immunostaining.
2, Although HA-tag staining is used, there is no quantification of actual PLEKHM1 overexpression level in vivo. It’s unclear if the observed effects are physiologically relevant or supra-physiological. The author should include data of PLEKHM1 mRNA/protein expression levels via qPCR or western blot from injected regions.
3, There is no behavioral or motor function assessment in their PD mouse model. Even simple tests (e.g., open field, rotarod) could provide valuable correlation between pathology and function. The authors are requested to incorporate behavioral testing of mice.
4, prior work shows that both loss and gain of PLEKHM1 can impair autophagy. The study focuses only on overexpression. In this study there’s no loss-of-function or rescue experiment to distinguish direct toxic gain vs. disrupted stoichiometry of autophagy machinery. The authors are suggested to consider or discuss a rescue experiment where PLEKHM1 overexpression is partially knocked down, or compare to a dominant-negative version.
5, The mechanistic insight of the current study is weak. The link between autophagy impairment and αSyn toxicity needs more mechanistic data.
minor defects:
1, Sample sizes are modest (n=5–6 per group); although justified, this limits power in detecting subtler effects.
2, Most data are presented as normalized ratios (treated vs. contralateral), which can obscure absolute differences. The authors re suggested to provide raw values or absolute counts in supplementary figures.
Round 2
Reviewer 1 Report
Comments and Suggestions for Authors
Although some previously suggested experiments and analysis would have improved the significance and accuracy of this study, this reviewer is overall satisfied with the comments provided by the authors.
Author Response
Thank you for your review and comments.
Reviewer 2 Report
Comments and Suggestions for Authors
Unfortunately, the authors’ revision responses remain insufficient, and the critical flaws have not been addressed. Based on the current data, I recommend rejection. The reasons are as follows:
- The author might not understand my question. Of course, I know the author conducted HA staining to assess the exogenously expressing level of PLEKHM1 but did not evaluate the "baseline" endogenous expression of PLEKHM1. To address this, please perform in situ hybridization to determine the native expression levels of PLEKHM1 and quantify the degree of upregulation in your system.
- The authors might have overlooked my point: this representative image itself is fundamentally flawed. It makes little sense to quantify only RFP+/GFP+ and GFP+/GFP– puncta without proper understanding and the optimal microscopy imaging. If RFP+/GFP+ puncta truly represent autophagosomes, why do they mainly colocalize with LAMP1? Where are the example puncta shown in Figure 4I? What is the meaning of checking colocalization of RFP+ with LAMP1? How is this shift explained in PLEKHM1’s function? The authors must first confirm that the RFP-EGFP-LC3 reporter mouse is functioning correctly as an autophagy reporter in this tissue and your experiments. If it is not, they should perform additional assays to pinpoint the effect of PLEKHM1 on autophagy.
Author Response
The author might not understand my question. Of course, I know the author conducted HA staining to assess the exogenously expressing level of PLEKHM1 but did not evaluate the "baseline" endogenous expression of PLEKHM1. To address this, please perform in situ hybridization to determine the native expression levels of PLEKHM1 and quantify the degree of upregulation in your system.
Throughout the study, we used an rAAV-control vector (rAAV-EGFP), which ensured that any potential effects on endogenous PLEKHM1 expression were adequately controlled. We therefore believe that performing in situ hybridization to determine baseline expression is outside the scope of this work. PLEKHM1 transgene expression was confirmed in all experimental animals and was present in a substantial number of cells within the SN (see Figure 1B). As is well established, AAV- and lentiviral-mediated gene delivery typically results in supraphysiological expression levels. We assume this is also the case in our study and consider this a well-recognized characteristic of such gene delivery approaches that the intended audience will be familiar with.
The authors might have overlooked my point: this representative image itself is fundamentally flawed. It makes little sense to quantify only RFP+/GFP+ and GFP+/GFP– puncta without proper understanding and the optimal microscopy imaging. If RFP+/GFP+ puncta truly represent autophagosomes, why do they mainly colocalize with LAMP1? Where are the example puncta shown in Figure 4I? What is the meaning of checking colocalization of RFP+ with LAMP1? How is this shift explained in PLEKHM1’s function? The authors must first confirm that the RFP-EGFP-LC3 reporter mouse is functioning correctly as an autophagy reporter in this tissue and your experiments. If it is not, they should perform additional assays to pinpoint the effect of PLEKHM1 on autophagy.
We performed confocal image acquisition for these analyses. Figure 4I represents the ratio of the area covered by autolysosomal markers to the area marked by autophagosomal markers, rather than a count of individual puncta.
Regarding the colocalization of RFP+ with LAMP1, the shift could be explained—together with the reduced LAMP1-positive area—by a failure of lysosomal recycling. This remains a working hypothesis, which we discuss openly in the Discussion section.
We agree that correct functioning of the RFP-EGFP-LC3 reporter in this tissue is essential for interpretation. This reporter has been validated in multiple prior studies and in similar tissue contexts, and we have no indication of improper reporter function in our experimental setting. One limitation, which we already acknowledge, is the absence of a control injection in RFP-EGFP-LC3 mice, as these animals already express EGFP. Nonetheless, our experimental treatments produced changes in widely used autophagy readouts. Crucially, the new data presented in Figure 5 incorporates an additional control (untreated hemisphere and rAAV-EGFP).
To further address your concern and to better assess the functional consequences of the observed changes in the pathway, we performed additional analyses using p62 as an independent marker. Our new data indicate that PLEKHM1 overexpression compromises autophagic flux. These results are now included in the revised manuscript (see new Figure 5) and discussed in greater detail in the expanded Discussion section.

Reviewer 3 Report
Comments and Suggestions for Authors
The authors have responded appropriately to all questions I raised. I suggest the editor may consider accepting this manuscript to be published in its current state.
Author Response
Thank you for your review and comments.
Reviewer 4 Report
Comments and Suggestions for Authors
Although the authors have revised the article and provided some new data, all findings are correlative. It is just confirmed that virus expression of PLEKHM1 induce more cell death in the presence of synuclein. However it is not clear how these events are linked together. Furthermore the study can not provide evidence of protein expression of syn and PLEKHM1. some key autophagy markers are also not studied. So the scientific information provided from the current study is very limited, even the in vivo mice model is investigated. So I suggest rejection of this study.
Author Response
Although the authors have revised the article and provided some new data, all findings are correlative. It is just confirmed that virus expression of PLEKHM1 induce more cell death in the presence of synuclein. However it is not clear how these events are linked together. Furthermore the study can not provide evidence of protein expression of syn and PLEKHM1. some key autophagy markers are also not studied. So the scientific information provided from the current study is very limited, even the in vivo mice model is investigated. So I suggest rejection of this study.
Contrary to the reviewer’s statement, our study does provide evidence of PLEKHM1, α-synuclein (including phosphorylated α-synuclein), and EGFP (control vector) expression, as shown in Figures 1B–E. We have performed multiple analyses proposing mechanisms of autophagy impairment as a potential underlying cause of the accelerated neurodegeneration.
In the revised manuscript, we have additionally conducted new analyses using p62 as a marker, which further suggest compromised autophagic flux as a possible mechanism linking PLEKHM1 overexpression and increased cell vulnerability in the presence of α-synuclein.
We also continue to clearly acknowledge the limitations of our study. Nonetheless, we believe the results provide meaningful in vivo evidence supporting the link between PLEKHM1 as a risk factor and α-synuclein–related pathology.
